# Imaging Changes and Immune-Checkpoint Expression on T Cells in Bronchoalveolar Lavage Fluid from Patients with Pulmonary Sarcoidosis

**DOI:** 10.3390/biomedicines9091231

**Published:** 2021-09-16

**Authors:** Yasuaki Kotetsu, Toyoshi Yanagihara, Kunihiro Suzuki, Hiroyuki Ando, Daisuke Eto, Kentaro Hata, Masako Arimura-Omori, Yuzo Yamamoto, Eiji Harada, Naoki Hamada

**Affiliations:** 1Research Institute for Diseases of the Chest, Graduate School of Medical Sciences, Kyushu University, Fukuoka 812-8582, Japan; kotetsu.yasuaki.487@m.kyushu-u.ac.jp (Y.K.); suzuki.kunihiro.784@m.kyushu-u.ac.jp (K.S.); ando.hiroyuki.781@m.kyushu-u.ac.jp (H.A.); eto.daisuke.925@m.kyushu-u.ac.jp (D.E.); hata.kentaro.369@m.kyushu-u.ac.jp (K.H.); masako-o@med.kyushu-u.ac.jp (M.A.-O.); yamamoto.yuzo.669@m.kyushu-u.ac.jp (Y.Y.); harada.eiji.827@m.kyushu-u.ac.jp (E.H.); hamada.naoki.608@m.kyushu-u.ac.jp (N.H.); 2Department of Respiratory Medicine, Hamanomachi Hospital, Fukuoka 810-8539, Japan

**Keywords:** immune checkpoint, sarcoidosis, bronchoalveolar lavage

## Abstract

Sarcoidosis is a systemic, granulomatous disease caused by unknown immunological abnormalities. The organs most vulnerable to sarcoidosis are the lungs. Patients often resolve spontaneously, but the lungs can also be severely affected. Although details regarding prognostic factors in sarcoidosis patients with lung involvement remain unclear, several reports have suggested that immune checkpoint molecules are involved in the pathogenesis of sarcoidosis. In this study, we divided sarcoidosis patients into two groups based on chest computed tomography (CT) findings and compared immune checkpoint molecules expressed on T cells in bronchoalveolar lavage fluid (BALF) in the two groups, using flow cytometry. We found elevated programmed cell death 1 (PD-1) or T cell immunoglobulin- and mucin-domain-containing molecule-3 (TIM-3) expression on T cells in BALF in patients with spontaneous improvement in CT findings, compared with those in patients without improvement in CT findings. In conclusion, our study implies that PD-1 or TIM-3 expression on T cells in BALF may be a prognostic factor for pulmonary lesions in sarcoidosis.

## 1. Introduction

Sarcoidosis is a granulomatous disease that can develop in almost any organ, but the organs most vulnerable to sarcoidosis are the lungs [1]. Half of patients resolve spontaneously within two years, and many other patients do so within five years. On the other hand, some patients may be in mortal danger if the disease becomes chronic and causes pulmonary fibrosis [2].

Sarcoidosis has been known for a long time, but a detailed understanding of factors leading to spontaneous resolution of the imaging findings of this disease does not yet exist. It has been reported that restoration of CD4^+^ subset function is associated with spontaneous clinical resolution of pulmonary sarcoidosis [3]. T cell activation is regulated by co-stimulatory factors, such as CD28, and co-inhibitory factors, such as immune checkpoint molecules. When T cell receptor binding to antigen/major histocompatibility complex (MHC) is accompanied by the involvement of immune checkpoint molecules, T cell activation is suppressed [4]. We hypothesize that these immune checkpoint molecules may also participate in the pathogenesis of sarcoidosis and that expression of these molecules on T cells in the affected organ (lung) may influence the prognosis of sarcoidosis. We focus on the following proteins: programmed cell death 1 (PD-1), a negative regulator of activated T cells, and its ligand, programmed cell death 1 ligand 1 (PD-L1), both of which are targets of immunotherapy [5]. T cell immunoglobulin- and mucin-domain-containing molecule-3 (TIM-3), T cell immunoglobulin and ITIM domain (TIGIT), and lymphocyte activation gene 3 (LAG-3), the next generation of immune checkpoint molecules, also suppressively regulate immune responses of T cells through their unique signals [6]. 

High mobility group box 1 (HMGB1) is one of the ligands for TIM-3 [6]. Hamada et al. have reported that patients with idiopathic pulmonary fibrosis (IPF) have significantly higher levels of HMGB1 in BALF, compared with healthy controls [7]. Suchankova et al. have reported that patients with sarcoidosis have higher levels of HMGB1 in BALF, compared with patients with IPF [8]. On the other hand, the relationship between HMGB1 concentration and TIM-3 expressed on T cells in BALF has not been reported so far. If there is a correlation between them, it may be useful in predicting the prognosis of pulmonary sarcoidosis.

The purpose of this study was to investigate the relationship between immune checkpoint molecules expressed on T cells in bronchoalveolar lavage fluid (BALF) and gradual changes in imaging findings. To achieve this aim, we divided patients into two groups: one with improved computed tomography (CT) findings at follow-up (3 to 9 months after the first visit) and the other without. We retrospectively analyzed whether there were differences in the expression of immune checkpoint molecules on T cells in BALF in the two groups.

## 2. Materials and Methods

### 2.1. Patients

Patients who were subjected to BALF collection and newly diagnosed with sarcoidosis at Kyushu University Hospital between March 2017 and August 2018 were eligible for enrollment in this study. Diagnostic and radiographic criteria used to define sarcoidosis were applied [9,10]. The study was authorized by the Ethics Committee of Kyushu University Hospital (reference number 28–233), and written informed consent was obtained from all enrolled patients. BALF was collected as follows: A bronchoscope was guided into the subsegment of the lung that was to undergo BAL and advanced until the tip was wedged into a bronchiole. Sterile normal saline (50 mL) was injected with a handheld syringe, followed by a gradual withdrawing of the saline back into the syringe, repeated three times.

### 2.2. Criteria for CT Imaging Changes

Evaluation of CT imaging changes was performed for the target regions. Target regions here referred to lesions in the lung area, where BAL was performed. The nature of target lesions (granular, nodular, ground-glass opacity (GGO), consolidation) was irrelevant. Patients were considered to have improved if target lesions had decreased in size or disappeared at follow-up, compared to the first visit, or if the shadow was faded in the case of GGO or consolidation. Patients who did not meet the above criteria were classified as unimproved. Two respiratory physicians independently performed the assessment, and a decision of improved or unimproved was rendered when the two judgments agreed. In the case of disagreement, they discussed the results. If they then agreed, the case was included in subsequent analysis; if there was no consensus after discussion, the patient was excluded from the analysis.

### 2.3. Flow Cytometry

Flow cytometry was performed as described previously [11]. Briefly, BALF was filtered through a mesh to remove debris and centrifuged to isolate cells. The collected cells were first incubated on ice with Human BD Fc Block (#564220) (BD Biosciences, San Jose, CA, USA) for 10 min and then stained with appropriate antibodies for 20 min for analysis with a FACS Verse flow cytometer (BD Biosciences, San Jose, CA, USA). The following antibodies were used: Brilliant Blue 515-conjugated anti-human CD3 (#564465), peridinin chlorophyll protein complex (PerCP)- and cyanine 5.5 (Cy5.5)-conjugated anti-human CD4 (#560650), and phycoerythrin (PE)- and Cy7-conjugated anti-human CD8 (#335787) from BD Biosciences, San Jose, CA, USA; PE-conjugated anti-human PD-1 (#12-9969-42) from Thermo Fisher Scientific, Waltham, MA, USA; APC-conjugated anti-human TIM-3 (#345011), APC- conjugated anti-human TIGIT (#372706), APC-conjugated anti-human PD-L1 (#329708), PE-conjugated mouse immunoglobulin G1 (IgG1) κ isotype control (#400113) and APC-conjugated mouse IgG1 κ isotype control (#400119) from BioLegend, San Diego, CA, USA; and ATTO 647N-conjugated anti-human LAG-3 (ALX-804-806TS-T100) from Enzo Life Sciences, Farmingdale, NY, USA.

### 2.4. Immunofluorescence

Diagnostic formalin-fixed, paraffin-embedded tissue blocks obtained from patients with sarcoidosis were provided by Kyushu University Hospital. Paraffin sections (5 μm in thickness) were deparaffinized and boiled with ImmunoSaver (#097-06192, FUJIFILM Wako Pure Chemical Corporation, Osaka, Japan) for 45 min. After blocking with 5% BSA (#A2153, Sigma-Aldrich Co., St. Louis, MO, USA)/PBS for 30 min at room temperature, sections were incubated with primary antibodies overnight in a humidified chamber at 4 °C. After incubation with secondary antibody for 1 h at room temperature, sections were mounted in VECTASHIELD Mounting Medium with DAPI (#H-1200, Vector Laboratories, Burlingame, CA, USA). The sections were analyzed, using an LSM700 (Carl Zeiss, Oberkochen, Germany). The following antibodies were used: Rat anti-human CD3 (#ab11089) and Alexa Flour 555-conjugated donkey anti-rat IgG (#ab150154) from Abcam, Cambridge, UK; Rabbit anti-human PD-1 (#86163T), Rabbit anti-human TIM-3 (#45208T), Rabbit monoclonal IgG isotype control (#3900S), and Alexa Flour 488-conjugated goat anti-rabbit IgG (#4412S) from Cell Signaling Technology, Danvers, MA, USA.

### 2.5. ELISA

HMGB1 concentration in BALF was measured with an ELISA kit (Shino-Test Corporation, Sagamihara, Kanagawa, Japan).

### 2.6. Statistical Analysis

Statistical analysis was performed with the Mann–Whitney U test, using GraphPad Prism 9 (GraphPad Software Inc., San Diego, CA, USA). A *p*-value < 0.05 was considered statistically significant.

## 3. Results

### 3.1. Patient Characteristics

We enrolled 35 patients with suspected sarcoidosis at the first visit. In subsequent tests, 23 patients were diagnosed with sarcoidosis, 19 with a histological diagnosis, and the remaining 4 with a clinical diagnosis. We analyzed 11 patients who underwent follow-up with CT for background factors and BALF fractions at their first visit (Table 1). Of these sarcoidosis patients, 11 underwent follow-up with CT after 3 to 9 months. Six improved (Figure 1A), while the other five remained unchanged or worsened (Figure 1B). All patients were clinically asymptomatic or mildly ill, and did not receive systemic steroids, nor did they require them during follow-up. Imaging findings were evaluated, using Scadding stages, which are widely used classifications for intrathoracic lesions in pulmonary sarcoidosis, reported by John Scadding [10]. There was no significant difference in the Scadding stages between improved and unimproved groups (*p* = 0.83). There was also no significant difference in the proportion of lymphocytes or in the CD4/8 ratio in BALF in the two groups (*p* > 0.99 and 0.57, respectively).

### 3.2. PD-1 Expression on CD4^+^ T Cells Was Higher in the Improved Group

Previous reports have suggested that PD-1 and TIM-3 expressed on CD4^+^ T cells in BALF may be involved in the pathogenesis of sarcoidosis [12,13]. However, we analyzed the expression levels of PD-1 on CD4^+^ T and CD8^+^ T cells in BALF and found that the percentage of PD-1^+^ cells among CD4^+^ T cells was significantly higher for the improved group than for the unimproved group, with median interquartile range (IQR) values of 13.0% (9.2–19.1%) for the improved group, compared with 3.7% (2.6% to 7.3%) for the unimproved group (Figure 2A,B). On the other hand, there was no significant difference in the percentage of PD-1^+^ cells among CD8^+^ T cells in the two groups (Figure 2D,E).

### 3.3. TIM-3 Expression on CD4^+^ or CD8^+^ T Cells Was Higher in the Improved Group

We also examined the expression level of TIM-3 on CD4^+^ T cells and CD8^+^ T cells in BALF. The proportion of TIM-3^+^ cells among CD4^+^ T cells was significantly higher for the improved group than for the unimproved group, with median (IQR) values of 6.3% (3.0–9.0%) for the improved group, compared with 2.4% (2.1–2.7%) for the unimproved group (Figure 2A,C). Similarly, the proportion of TIM-3^+^ cells among CD8^+^ T cells was significantly higher for the improved group than for the unimproved group, with median (IQR) values of 11.1% (7.7–17.8%) for the improved group, compared with 4.8% (4.0–6.5%) for the unimproved group (Figure 2D,F).

### 3.4. TIGIT, LAG-3 and PD-L1 Expression on T Cells from Sarcoidosis Patients

We analyzed expression of TIGIT, LAG-3, and PD-L1 on T cells in BALF, and found no difference in expression between the two groups (Figure 3).

### 3.5. T Cells in Lung Specimens of Patients with Sarcoidosis Express PD-1 or TIM-3

The pathological features of sarcoidosis include T cell infiltration as well as non-caseating epithelioid cell granulomas [1]. We performed immunofluorescence staining of lung specimens from patients with pulmonary sarcoidosis to analyze whether T cells express PD-1 or TIM-3 not only in BALF, but also in lung specimens. We clarified that PD-1^+^ CD3^+^ T cells and TIM-3^+^ CD3^+^ T cells were present in lung tissue (Figure 4). However, the number of each was very small, and we could not determine their localization or whether there was a difference between the improved and unimproved groups.

### 3.6. Correlation of HMGB1 Concentration with TIM-3^+^ T Cells

Since we found that there was a significant difference in the expression of TIM-3 on T cells in BALF between the improved and unimproved groups (Figure 2), we focused on HMGB1, one of the ligands for TIM-3 [6]. We measured the concentration of HMGB1 in BALF and found that the concentration of HMGB1 tended to be higher in improved patients than in unimproved patients, although the difference was not significant. Furthermore, we analyzed the correlation between the concentration of HMGB1 and TIM-3^+^ T cells and found that there was no specific finding in the unimproved group, but there was an inverse correlation between the concentration of HMGB1 and the proportion of TIM-3^+^ T cells among CD8^+^ T cells in the improved group (Figure 5).

## 4. Discussion

The purpose of this study was to analyze whether there is an association between immune checkpoint expression on T cells in BALF and subsequent imaging changes in pulmonary sarcoidosis. We found that CD4^+^ T cells had higher expression of PD-1 or TIM-3 and that CD8^+^ T cells had higher expression of TIM-3 in the improved group, compared with the unimproved group. 

It has been reported that the expression of PD-1 on CD4^+^ T cells in BALF is higher in patients with sarcoidosis than in healthy controls [12], but it has been uncertain whether differences in PD-1 expression level are associated with changes in imaging findings. In this study, we found that patients with higher PD-1 expression on CD4^+^ T cells exhibited spontaneous resolution of imaging findings (Figure 1 and Figure 2A,B). However, PD-1 expression on CD8^+^ T cells was not significantly different between the two groups (Figure 2D,E). Unfortunately, it is difficult to determine from this experiment alone whether PD-1 expression on CD8^+^ T cells in the unimproved group splits into high and low expression, or whether there are individuals with median values who are inherently low but are high as outliers. This issue needs to be re-examined after increasing the number of cases.

Although it has been reported that expression of TIM-3 on CD4^+^ T cells in BALF is lower in patients with sarcoidosis, compared with healthy controls [13], the relationship between TIM-3 and sarcoidosis remains unclear. In this study, we found that higher expression of TIM-3 on CD4^+^ or CD8^+^ T cells was correlated with spontaneous remission of imaging findings. Given that both PD-1 and TIM-3 can negatively regulate T cell responses [6,14], higher expression of PD-1 or TIM-3 on CD4^+^ and CD8^+^ T cells may negatively regulate Th-1 cytokinesis and decrease mitotic activity so as to induce apoptosis, resulting in resolution of the granuloma and improved imaging findings.

Data from peripheral blood of patients with sarcoidosis showed that expression of PD-1 on CD4^+^ T cells, which was elevated at the time of exacerbation, improved to the same level as healthy controls after the clinical findings improved [12]. These data suggest that the inflammatory response of sarcoidosis may be terminated, due to suppressive regulation by T cell exhaustion, and that these cells may eventually be eliminated by apoptosis. It has also been reported that the proliferative capacity of CD4^+^ T cells in peripheral blood of patients with sarcoidosis was decreased, compared with healthy controls, and that proliferative capacity was restored to the level of healthy controls by blockade of the PD-1 pathway [12]. Additionally, case reports of new-onset of sarcoidosis, exacerbation, and sarcoidosis-like reactions after PD-1 blockade in patients with malignancies have been accumulating [15,16,17,18,19]. 

Though it is generally accepted that Th1 cells are involved in the pathogenesis of sarcoidosis [1], Lomax et al. proposed that sarcoidosis may develop when Th17.1 cells, which co-produce IFN-γ and IL-17, are amplified by anti-PD-1 immunotherapy [16]. PD-1 blockade is useful in anti-tumor immunity, but from case reports that PD-1 blockers caused sarcoidosis and sarcoidosis-like reactions, we speculated that blocking the PD-1 pathway may be disadvantageous for sarcoidosis. In addition, a close reading of the article by Braun et al. shows that % CD4^+^ T-cell proliferation of sarcoidosis patients varies greatly, compared with that of healthy subjects [12]. Although Braun et al. did not divide sarcoidosis patients by subsequent prognosis, our results suggest that patients with low % CD4^+^ T-cell proliferation may improve, while patients with high % CD4^+^ T-cell proliferation may remain unchanged or worsen. In summary, it is assumed that the role of T cells that accumulate in the lungs, whether they are Th1 cells or Th17.1 cells, is pathogenic, and that PD-1 suppresses the activation of these T cells. As described by Gkiozos et al., the analysis of ICI-induced sarcoid-like reactions may provide clues to the pathogenesis of sarcoidosis [20].

Previous reports have shown that CD4^+^ T cells in BALF express PD-1 and TIM-3, but we could not find any reports of such T cells in lung tissue. One of the pathological findings of sarcoidosis is infiltration of CD4^+^ T cells into the granuloma and infiltration of CD8^+^ T cells around the granuloma [21]. We performed immunofluorescence staining to determine whether T cells expressing PD-1 and TIM-3 were also present in the lung, and if so, whether their distribution was unique. We found PD-1^+^ CD3^+^ T cells and TIM-3^+^ CD3^+^ T cells in the lung tissue (Figure 4), but the number of cells was very small, and we could not characterize their distribution.

HMGB1 is one of the damage-associated molecular patterns (DAMPs), which has cytokine, chemokine, and growth factor activities by binding to other factors, and which regulates inflammatory and immune responses [22]. HMGB1 has been reported as a ligand for TIM-3 expressed by tumor-infiltrating DCs [23], and it has recently been suggested that the binding of HMGB1 to TIM-3 on regulatory CD8^+^ T cells can inhibit the expansion of effector T cells [24,25]. In this study, we demonstrated that the concentration of HMGB1 in BALF tends to be higher in the improved group than in the unimproved group (Figure 5A). 

We also demonstrated that the proportion of TIM-3^+^ cells among CD8^+^ T cells was inversely correlated with the concentration of HMGB1 in the improved group (Figure 5C). There are two mechanisms of HMGB1 release: passive release by cellular necrosis and active release from activated immune cells, including macrophages [26,27,28]. If HMGB1 is released by macrophages in the granuloma, HMGB1 will cause T cells to proliferate and release inflammatory cytokines in the early stage of inflammation, further activating macrophages. When the inflammation is prolonged and T cells are exhausted, they express TIM-3, which stops the release of cytokines and inhibits proliferation [29]. As a result, the activation of macrophages is also suppressed, the HMGB1 concentration decreases, and, finally, TIM-3^+^ T cells may persist as a remnant of inflammation. To confirm this hypothesis, it is necessary to analyze not only T cells, but also macrophages in the future.

We observed no significant differences in TIGIT, LAG-3, or PD-L1 expression on T cells in these two groups. Based on the results of the present experiments, we assume that TIGIT, LAG-3, and PD-L1 molecules on T cells in BALF may not contribute greatly to the regulation of the disease.

We acknowledge several limitations in this study. First, the degree of expression of immune checkpoint molecules is ambiguous. In actual clinical practice, it should be determined whether the level of immune checkpoint expression on T cells at the time of bronchoscopic examination is high or low, but in our experiment, we were not able to ensure a sufficient number of patients to establish a definite cutoff value. The incidence of sarcoidosis in Japan is low, at 1.01 per 100,000 population [30], and it is difficult to increase patient numbers at a single institution. In the future, it will be necessary to accumulate a larger number of cases in multicenter studies. 

Second, BAL could not be performed at the time of CT follow-up. Although it is important to analyze whether there are changes in immune checkpoint molecules expressed on T cells in BALF when there are changes in imaging findings, for ethical reasons, it is difficult to perform BAL procedures on patients who show no deterioration in imaging findings. We will continue to accumulate cases and perform BAL when imaging findings are markedly worsened and will compare the findings with those of BALF when patients are stable. 

Third, we could not quantify and compare the number of PD-1^+^ CD3^+^ T cells and TIM-3^+^ CD3^+^ T cells from immunofluorescence staining of lung specimens between the two groups. The samples were from transbronchial lung biopsies, so they were too small to count enough cells. Further, since false positives are often a problem with immunofluorescent staining, we used a negative control for comparison and a strict gain setting. As a result, sensitivity for PD-1^+^ and TIM-3^+^ may have been very poor, and some positive cells may have been classified as negative. Given the results of flow cytometry, we expect that intensity levels of PD-1 and TIM-3 are very low compared to cell specific markers such as CD3. Therefore, we were only able to verify the presence of PD-1^+^ CD3^+^ T cells and TIM-3^+^ CD3^+^ T cells in lung tissue. 

## 5. Conclusions

From the results of this study, although a detailed mechanism is yet unknown, high expression of PD-1 or TIM-3 on T cells in BALF appears to promote spontaneous remission and should be carefully investigated.

## Figures and Tables

**Figure 1 biomedicines-09-01231-f001:**
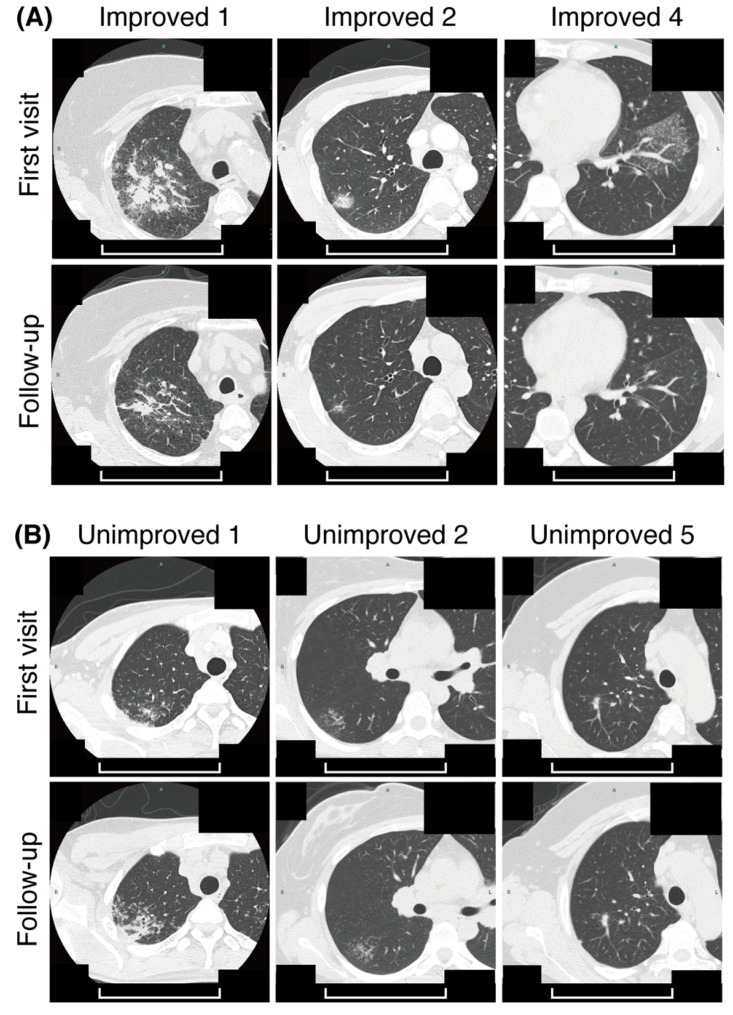
Computed tomography (CT) imaging changes in patients with sarcoidosis. CT images of three representative patients from the improved group (**A**) and three more from the unimproved group (**B**) at the time of their first visit (upper panels) and at the time of follow-up (after 3 to 9 months) (lower panels) are shown. Scale bars: 10 cm.

**Figure 2 biomedicines-09-01231-f002:**
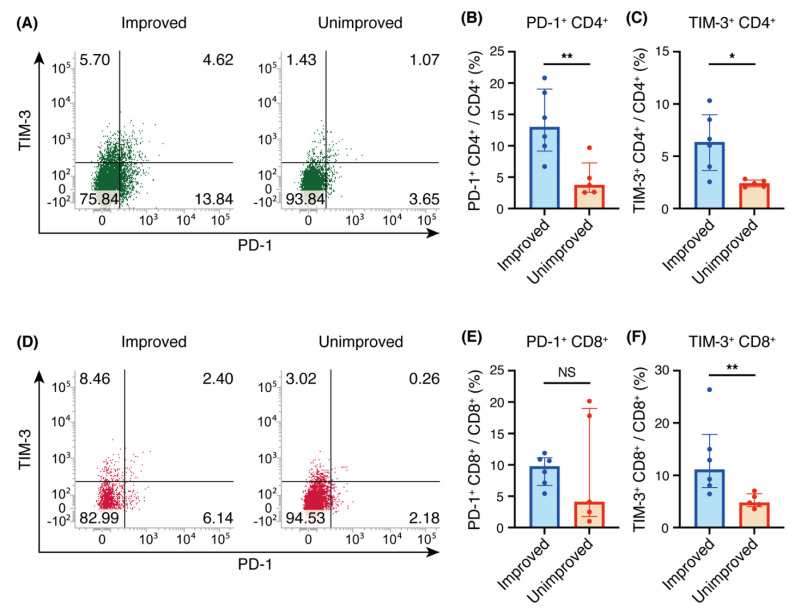
Programmed cell death 1 (PD-1) and T-cell immunoglobulin- and mucin-domain-containing molecule-3 (TIM-3) expression on CD4^+^ and CD8^+^ T cells in bronchoalveolar lavage fluid (BALF). (**A**,**D**) Representative flow cytometric analysis of PD-1 and TIM-3 expression on CD4^+^ T cells (**A**) and CD8^+^ T cells (**D**) in BALF. (**B**,**E**) Percentages of PD-1^+^ cells among CD4^+^ T cells (**B**) and CD8^+^ T cells (**E**). (**C**,**F**) Percentages of TIM-3^+^ cells among CD4^+^ T cells (**C**) and CD8^+^ T cells (**F**). (**B**,**C**,**E**,**F**) Individual patient data as well as medians and interquartile ranges are shown. * *p* < 0.05; ** *p* < 0.01 (Mann-Whitney U test).

**Figure 3 biomedicines-09-01231-f003:**
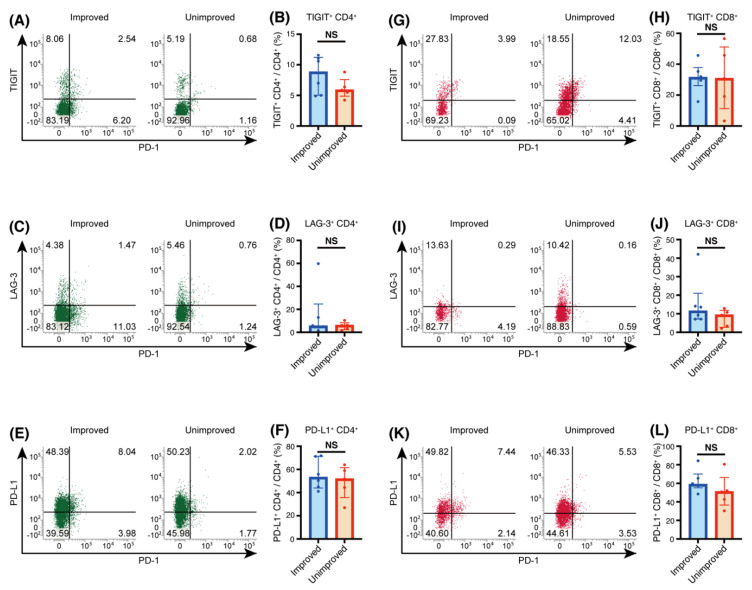
T cell immunoglobulin and ITIM domain (TIGIT), lymphocyte activation gene 3 (LAG-3), and programmed cell death 1 ligand 1 (PD-L1) expression on CD4^+^ and CD8^+^ T cells in bronchoalveolar lavage fluid (BALF). (**A**,**G**) Representative flow cytometric analysis of TIGIT expression on CD4^+^ T cells (**A**) and CD8^+^ T cells (**G**) in BALF. (**C**,**I**) Representative flow cytometric analysis of LAG-3 expression on CD4^+^ T cells (**C**) and CD8^+^ T cells (**I**) in BALF. (**E**,**K**) Representative flow cytometric analysis of PD-L1 expression on CD4^+^ T cells (**E**) and CD8^+^ T cells (**K**) in BALF. (**B**,**H**) Percentages of TIGIT^+^ cells among CD4^+^ T cells (**B**) and CD8^+^ T cells (**H**). (**D**,**J**) Percentages of LAG-3^+^ cells among CD4^+^ T cells (**D**) and CD8^+^ T cells (**J**). (**F**,**L**) Percentages of PD-L1^+^ cells among CD4^+^ T cells (**F**) and CD8^+^ T cells (**L**). (**B**,**D**,**F**,**H**,**J**,**L**) Individual patient data as well as medians and interquartile ranges are shown. NS: not significant.

**Figure 4 biomedicines-09-01231-f004:**
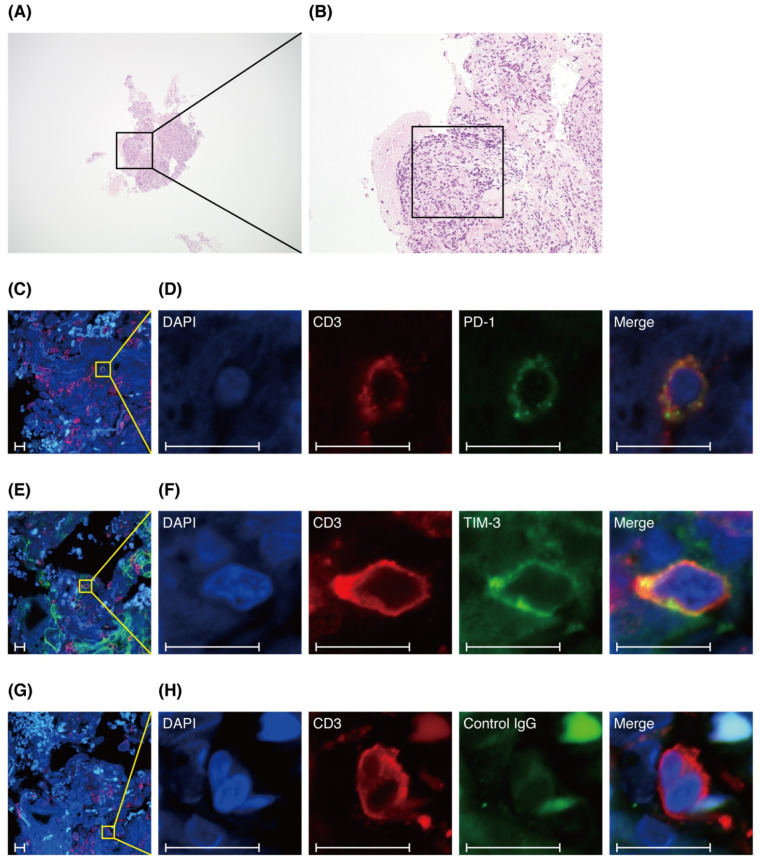
Programmed cell death 1 (PD-1) and T-cell immunoglobulin- and mucin-domain-containing molecule-3 (TIM-3) expression on CD3^+^ T cells in lung specimens. Hematoxylin–eosin (HE) stained 4× image (**A**) and 20× image (**B**). The square in (**B**) indicates the location of the immunofluorescence staining analysis. (**C**–**H**) Immunofluorescence staining of DAPI (blue), CD3 (red), PD-1 (green), TIM-3 (green), and Control IgG (green). (**C**,**E**,**G**) Images of 40×. (**D**,**F**,**H**) crop figures, respectively. Scale bars: 10 μm.

**Figure 5 biomedicines-09-01231-f005:**
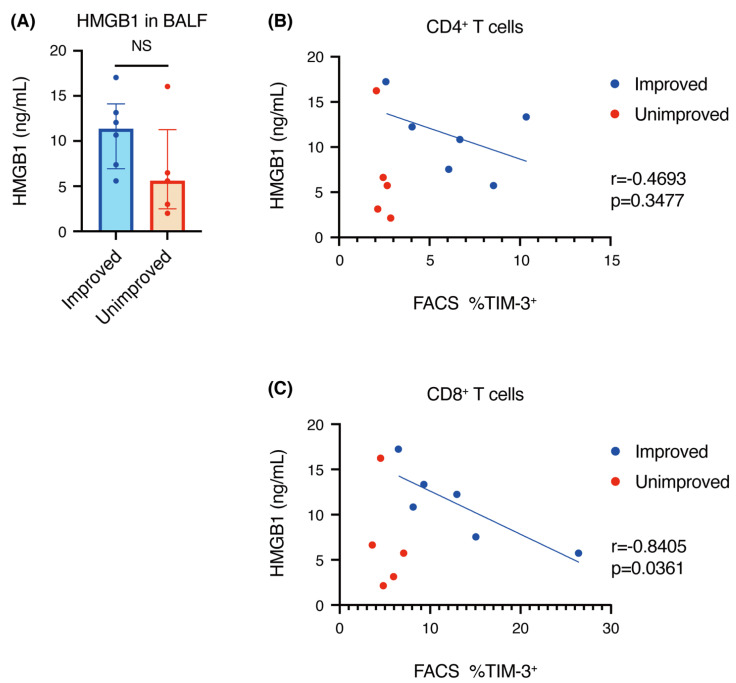
Correlation of the concentration of high mobility group box 1 (HMGB1) with the proportion of T-cell immunoglobulin- and mucin-domain-containing molecule-3 (TIM-3) positive T cells among CD4^+^ or CD8^+^ T cells in bronchoalveolar lavage fluid (BALF). (**A**) The concentration of HMGB1 in BALF was measured by ELISA. Individual patient data as well as medians and interquartile ranges are shown. (**B**,**C**) Correlation of the HMGB1 concentration with the proportion of TIM3^+^ cells among CD4^+^ T cells (**B**) or CD8^+^ T cells (**C**). Statistical analysis for Pearson’s correlation was performed using Student’s *t*-distribution. NS: not significant.

**Table 1 biomedicines-09-01231-t001:** Characteristics of patients and demographics of bronchoalveolar lavage fluid.

	Diagnosis	Sex	Scadding Stage	Extrapulmonary Disease	Systemic Steroid	Macrophages (%)	Neutrophils (%)	Lymphocytes (%)	Eosinophils (%)	CD4/CD8 Ratio
improved 1	Histological	F	2	Skin, Muscle	No	75.1	3.6	20.2	1.1	4.5
improved 2	Histological	M	2	No	No	82.4	1	16.2	0.4	19.5
improved 3	Histological	F	1	Heart, Muscle	No	49.3	0.2	50.1	0.4	9.6
improved 4	Histological	M	2	Eyes	No	80.3	0.3	19.3	0.1	4.8
improved 5	Histological	F	0	Eyes, Skin	No	52.7	0.3	46.9	0.1	7.3
improved 6	Clinical	F	1	Eyes, Heart, Spleen	No	43.7	0.7	55.3	0.3	5
unimproved 1	Histological	F	2	Eyes, Nervous system	No	55.4	0.9	43.3	0.4	11.9
unimproved 2	Histological	F	2	Eyes	No	90.7	0.2	9.1	0	9.7
unimproved 3	Histological	F	1	Eyes, Parotid gland	No	56.9	0.4	41.7	1.9	4.8
unimproved 4	Histological	F	2	Eyes, Skin, Muscle	No	41.8	2.8	54.1	1.3	1.6
unimproved 5	Histological	F	1	Skin, Lymph nodes	No	51.1	0.1	48.5	0.2	1.3

## Data Availability

The analyzed data sets in this study are available from corresponding author on reasonable request.

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
