# Peer review of "Imaging Changes and Immune-Checkpoint Expression on T Cells in Bronchoalveolar Lavage Fluid from Patients with Pulmonary Sarcoidosis"

_biomedicines, 2021, doi:10.3390/biomedicines9091231_

Round 1
Reviewer 1 Report
The current research discussed the prognostic factors in sarcoidosis patients with lung involvement and suggested that immune checkpoint molecules could be involved in the pathogenesis of sarcoidosis. To this hypothesis, the authors selected the sarcoidosis patients into two groups based on chest computed tomography (CT) findings and compared immune checkpoint molecules expressed on T cells.
The study noted that elevated programmed cell death 1 (PD-1) or T- cell immunoglobulin- and mucin-domain-containing molecule-3 (TIM-3) expression on T cells in patients with spontaneous improvement in CT findings compared with those in patients without improvement in CT findings. The study concluded that PD-1 or TIM-3 expression on T cells could be a prognostic factor for pulmonary lesions in sarcoidosis.
Decision: Minor comments
Below are the comments for this paper to be incorporated in the revised version of the manuscript.
- Line 35. “Since immune checkpoint” There general points needed to add immune checkpoint molecules
- Line 48 mention the time period in this sentence
- Line 50-52 No need to add the result lines in the introduction
- Line 56 were used two times, delete the second one
- Line 62 Fifty mL need to be used in numerical and also line 99, Twenty-three should be 23
- Line 107 please explain the criteria “Scadding criteria”
- Line 124-125 Is there was any explanation for this observation
- Need to include a paragraph on HMGB1 in the introduction
- Line 191-192 is there was any reason or any kind of the previous report suggested the author's observation
- Line 214-217 These studies needs to be explored more in the paragraph
- Line 233-234 Figure 5c what does it indicate
- The discussion portion needs to be elaborate based on the previous findings and the results noted by the authors
- Please check reference 13 and modify based on the journal style
Author Response
Response to Reviewer 1 Comments
Point 1: Line 35. “Since immune checkpoint” There general points needed to add immune checkpoint molecules
Response 1: As suggested, we have added a short explanation about immune checkpoint molecules (Lines 35-39), as follows:
T cell activation is regulated by co-stimulatory factors such as CD28 and co-inhibitory factors such as immune checkpoint molecules. When T cell receptor binding to antigen/major histocompatibility complex (MHC) is accompanied by the involvement of immune checkpoint molecules, T cell activation is suppressed [4].
Point 2: Line 48 mention the time period in this sentence
Response 2: The time period (3 to 9 months after the first visit) has been added, following the reviewer's suggestion (Lines 59-60).
Point 3: Line 50-52 No need to add the result lines in the introduction
Response 3: We have removed this part from the Introduction.
Point 4: Line 56 were used two times, delete the second one
Response 4: We have revised the sentence (Lines 65-67) as follows:
Patients who were subjected to BALF collection and newly diagnosed with sarcoidosis at Kyushu University Hospital between March 2017 and August 2018 were eligible for enrollment in this study.
Point 5: Line 62 Fifty mL need to be used in numerical and also line 99, Twenty-three should be 23
Response 5: It is generally not a good idea to write small numerals at the beginning of sentences in English. We followed the reviewer’s suggestion and checked to be certain that we did not start sentences with numbers (Lines 72-74 and Lines 128-130).
Point 6: Line 107 please explain the criteria “Scadding criteria”
Response 6: Around 1950, John Scadding reported that disease classification of intrathoracic lesions in pulmonary sarcoidosis could be based on features of chest radiographs. Later, this disease classification was found to be useful for prognosis and became widely used (Reference 10). We have added a short explanation with a reference in the text (Lines 136-137).
Point 7: Line 124-125 Is there was any explanation for this observation
Response 7: We appreciate the reviewer’s comment. Unfortunately, based on the results of this experiment alone, it is impossible to determine whether PD-1 expression on CD8+ T cells in the unimproved group splits into high and low expression, or whether there are individuals with median values who are inherently low, but are high as outliers. As mentioned in the Limitation section, we need to increase the number of cases and re-examine this issue. This has been noted in the second paragraph of the Discussion section (Lines 225-230) as follows:
However, PD-1 expression on CD8+ T cells was not significantly different between the two groups (Figure 2D,E). Unfortunately, it is difficult to determine from this experiment alone whether PD-1 expression on CD8+ T cells in the unimproved group splits into high and low expression, or whether there are individuals with median values who are inherently low, but are high as outliers. This issue needs to be re-examined after increasing the number of cases.
Point 8: Need to include a paragraph on HMGB1 in the introduction
Response 8: We appreciate and agree with the reviewer's comment. Since it is necessary to mention why we focused on HMGB1 in the Introduction, we added the following sentences to the third paragraph of the introduction (Lines 48-55).
High mobility group box 1 (HMGB1) is one of the ligands for TIM-3 [6]. Hamada et al. have reported that patients with idiopathic pulmonary fibrosis (IPF) have significantly higher levels of HMGB1 in BALF compared with healthy controls [7]. Suchankova et al. have reported that patients with sarcoidosis have higher levels of HMGB1 in BALF compared with patients with IPF [8]. On the other hand, the relationship between HMGB1 concentration and TIM-3 expressed on T cells in BALF has not been reported so far. If there is a correlation between them, it may be useful in predicting the prognosis of pulmonary sarcoidosis.
Point 9: Line 191-192 is there was any reason or any kind of the previous report suggested the author's observation
Response 9: I apologize for the lack of explanation. Based on the results of the present experiments, we assume that TIGIT, LAG-3, and PD-L1 molecules on T cells in BALF may not contribute significantly to regulation of the disease. We could not find any reports that show TIGIT, LAG-3 and PD-L1 expression on T cells in BALF and in relation to the prognosis of the disease. We have added this explanation in the manuscript (Lines 291-294).
Point 10: Line 214-217 These studies needs to be explored more in the paragraph
Response 10: We appreciate the reviewer’s important recommendation. We have corrected our leap in logic. PD-1 blockers are useful in anti-tumor immunity, but from case reports that PD-1 inhibition causes sarcoidosis and sarcoidosis-like reactions, we speculated that blocking the PD-1 pathway may be disadvantageous in sarcoidosis. In addition, a close reading of the article by Braun et al. shows that the % CD4+ T-cell proliferation of sarcoidosis patients varies greatly compared with that of healthy subjects. Although they did not divide sarcoidosis patients by subsequent prognosis, our results suggest that those with low % CD4+ T-cell proliferation may improve in the future and those with high % CD4+ T-cell proliferation may worsen. In other words, the role of T cells in sarcoidosis is pathogenic, and PD-1 suppresses activation of these T cells. We have revised the text as follows (Lines 250-263):
Though it is generally accepted that Th1 cells are involved in the pathogenesis of sarcoidosis [1], Lomax et al. have proposed that sarcoidosis may develop when Th17.1 cells, which co-produce IFN-γ and IL-17, are amplified by anti-PD-1 immunotherapy [16]. PD-1 blockade is useful in anti-tumor immunity, but from the case reports that PD-1 blockers caused sarcoidosis and sarcoidosis-like reactions, we speculated that blocking the PD-1 pathway may be disadvantageous for sarcoidosis. In addition, a close reading of the article by Braun et al. shows that the % CD4+ T-cell proliferation of sarcoidosis patients varies greatly compared with that of healthy subjects [12]. Although Braun et al. did not divide sarcoidosis patients by subsequent prognosis, our results suggest that patients with low % CD4+ T-cell proliferation may improve and patients with high % CD4+ T-cell proliferation may remain unchanged or worsen. In summary, it is assumed that the role of T cells that accumulate in the lungs, whether they are Th1 cells or Th17.1 cells, is pathogenic, and that PD-1 suppresses activation of these T cells. As described by Gkiozos et al, the analysis of ICI-induced sarcoid-like reactions may provide clues to the pathogenesis of sarcoidosis [20].
Point 11: Line 233-234 Figure 5c what does it indicate
Response 11: We explained what Figure 5c indicates in the Discussion section (Lines 282-289).
Point 12: The discussion portion needs to be elaborate based on the previous findings and the results noted by the authors
Response 12: We sincerely appreciate the opportunity the reviewer has given us to review and re-think our paper through excellent questions. We have revised the Discussion section based on suggestions above.
Point 13: Please check reference 13 and modify based on the journal style
Response 13: We have corrected "Annals of Oncology" to "Ann. Oncol.", according to the MDPI Reference List and Citations Style Guide (Line 378).
Reviewer 2 Report
The manuscript entitled “Imaging changes and immune-checkpoint expression on T cells in bronchoalveolar lavage fluid from patients with pulmonary sarcoidosis” demonstrates that PD1 and TIM3 expression by CD4/CD8 T lymphocytes in bronchoalveolar lavage fluid (BALF) has a prognostic value in patients with sarcoidosis. The manuscript is well-written and the findings are interesting with a potential translational impact. The authors acknowledge the main limitation of this study which is the small number of the cohort. Overall, the manuscript is suitable for publication to Biomedicines. A few comments are provided below.
- The catalogue numbers of the antibodies should be added in the Material and Methods.
- Did the authors count the number of CD3+TIM3+ and CD3+PD1+ per field in the two subgroups of patients? It would be meaningful to provide this data.
Author Response
Point 1: The catalogue numbers of the antibodies should be added in the Material and Methods.
Response 1: We have replaced the clone names on the antibodies with the catalog numbers (Lines 93-105 and Lines 116-119). In addition, there was an error in the description of the anti-human PD-1 antibody, which is not a BioLegend product, but a Thermo Fisher Scientific product. We have corrected this error (Line 101).
Point 2: Did the authors count the number of CD3+TIM3+ and CD3+PD1+ per field in the two subgroups of patients? It would be meaningful to provide this data.
Response 2: We appreciate the reviewer’s instructive suggestion. We also thought that this point is very important. We had predicted that levels of PD-1 and TIM-3 expressed on T cells in lung tissue would differ between the two groups, as well as in BALF. We tried to count the number of PD-1+ CD3+ T cells and TIM-3+ CD3+ T cells in each sample, but there were fewer than 5 cells per sample and no significant difference between the two groups. First, the samples from trans-bronchial lung biopsy were too small to count enough cells. Further, since false positives are often a problem with immunofluorescent staining (unfortunately, there are some questionable papers on the subject), we used a negative control for comparison and a strict gain setting. As a result, the sensitivity of PD-1+ and TIM-3+ may have been very poor, and some positive cells may have been classified as negative. Given the results of flow cytometry, we expect that the intensity levels of PD-1 and TIM-3 are very low compared to cell-specific markers such as CD3. Therefore, we were only able to verify the presence of PD-1+ CD3+ T cells and TIM-3+ CD3+ T cells in lung tissue. We have added an explanation to the Discussion section (Lines 310-319).
Reviewer 3 Report
The Authors reported on the expression of immune-checkpoints molecules on the membrane of T cells retrieved from bronchoalveolar lavage fluid in patients affected with sarcoidosis of the lung. Elevated levels of programmed cell death 1 (PD-1) or T- 19 cell immunoglobulin- and mucin-domain-containing molecule-3 (TIM-3) expression were found in patients with spontaneous improvement in computed tomography findings compared to patients not showing any improvement. Authors hence claim that PD-1 or TIM-3 expression on T cells bronchoalveolar lavage fluid may be able to prognosticate in this setting.
- Authors subdivided pts according to computed tomography findings, stratifying in those who improved and those who did not. Amogst those who underwent compted tomogeaphy-based follow up (11), 6 improved and 5 got worse or remained stable. Can you please provide details about the computed-tomography based stratification? It reamins unclear to me how was it done. This is crucial, since it has an impact on the further analysis. Particularly because Scadding criteria were not different between the 2 groups.
- Authors observed that CD4+ T cells had higher expression of PD-1 or TIM- 189 3 and that CD8+ T cells had higher expression of TIM-3 in patients showing an improvement in cumputed tomography findings. Authors suggest these molecules may have been involved in the pathogenesis of sarcoidois, implying causality. How can we exclude it is just a spurious observation? Or this is just a surface membrane phenotype of T celles in reponders, with casality involving other patterns?
- Bronchoalveolar lavage fluid findinds during follow up are not reported. This is acknowledged as a limitation of the study.
Round 2
Reviewer 3 Report
I would like to ackowledge the authors for the changes made on the manuscript which comply with the suggestions provided. I do not have any further comment.